# Prioritisation of ocean biodiversity data collection to deliver a sustainable ocean
Amelia E. H. Bridges [1] ✉ & Kerry L. Howell[1,2]

Fundamental ecological questions about the distribution of ocean life remain unanswered, hindering both the effective management of the ocean, and our comprehension of life on this planet. The benthic and pelagic realms are subject to different methods of study, and to understand where to best focus effort, a thorough understanding of existing information is required, allowing identification of critical knowledge gaps. Open-access data repositories provide a valuable means to identify such gaps; however, these repositories are subject to challenges in separating benthic from pelagic data. Here we demonstrate an automated data pipeline for extracting and separating benthic from pelagic data in open-access repositories. By stratifying data against essential ocean variables in a critical gap analysis, we show that large spatial and taxonomic biases exist in both the benthic and pelagic global datasets, favouring depths shallower than ~100 m, the northern hemisphere, and vertebrate species. The newly compiled, cleaned, and classified dataset is used to identify areas of chronic under sampling and high-priority regions for exploration. We argue that coordinated strategic prioritisation of sampling is needed to support modelling and prediction, enabling us to better manage our oceans and comprehend life on Earth.

Exploration of marine ecosystems beyond recreational diver depths (30 m) poses a formidable challenge despite technological advancements[1], rendering these systems among the least studied on Earth[2]. The logistical difficulties inherent in reaching these depths have resulted in a scarcity of biological data, leaving fundamental questions unanswered regarding the diversity, distribution, and connectivity of life in the deep ocean[3,4]. Yet, whilst *exploration* of the deep ocean has accelerated over the last two centuries, so has *exploitation*, with the direct and indirect effects of anthropogenic activity being documented[5–7]. Not only does the absence of crucial information lessen the efficacy of sustainable management, but it also hampers our comprehension of various biological disciplines – how can we ascertain the universal principles governing life when confronted with a vast environment for which essential data are lacking?

Filling these data gaps for offshore benthic ecosystems aligns with the UN Decade of Ocean Science for Sustainable Development's vision to support '*the science we need for the ocean we want*' (A/RES/72/73), particularly their calls for research to advance understanding of deep-sea ecosystems. The role such data would play in strengthening ocean management and answering fundamental ecological questions also speaks to the UN's Sustainable Development Goal (SDG) 14 'Life Below Water' from 2015, and the more recent Kunming-Montreal 2030 Global Biodiversity Framework (GBF) targets – a successor to the 2011 Aichi Biodiversity Targets –

launched in 2023 (specifically, targets falling within the overarching global goals A and B; CBD/COP/15/L25). Facilitating the sustainable use of marine biodiversity is also central to the Agreement under the United Nations Convention on the Law of the Sea on the Conservation and Sustainable Use of Marine Biological Diversity of Areas beyond National Jurisdiction (A/CONF.232/2023/4), also known as the High Seas Treaty.

A dataset with the power to answer fundamental questions, quantify biodiversity, and characterize species ecological niches in the deep sea would need to be global in coverage, with equal stratification across key drivers of biodiversity such as latitude and depth[4]. Whilst latitude and depth are, themselves, biologically irrelevant, they serve as proxies for several other variables (e.g., pressure, temperature) and thus act as powerful predictors of biodiversity across ecosystems, including in the deep ocean[8–12]. Changes in both latitude and depth represent changes in energy availability and consequently diversity[13]. Thermal energy (i.e., temperature) typically decreases with increasing latitude and increasing depths, and chemical energy (e.g., particulate organic carbon (POC) flux to depth), decreases with increasing depth and displays a temperate peak – a more complex but well-defined relationship with latitude[14]. In addition to spanning extensive latitudinal and bathymetric ranges, data should be sourced from across the tree of life, enabling a comprehensive exploration of the distribution patterns of various phyla, life histories, and evolutionary strategies.

[1]School of Biological and Marine Sciences, University of Plymouth, Drake Circus, Plymouth, Devon, UK. [2]Plymouth Marine Laboratory, Prospect Pl, Plymouth, UK.
✉e-mail: amelia.bridges@plymouth.ac.uk

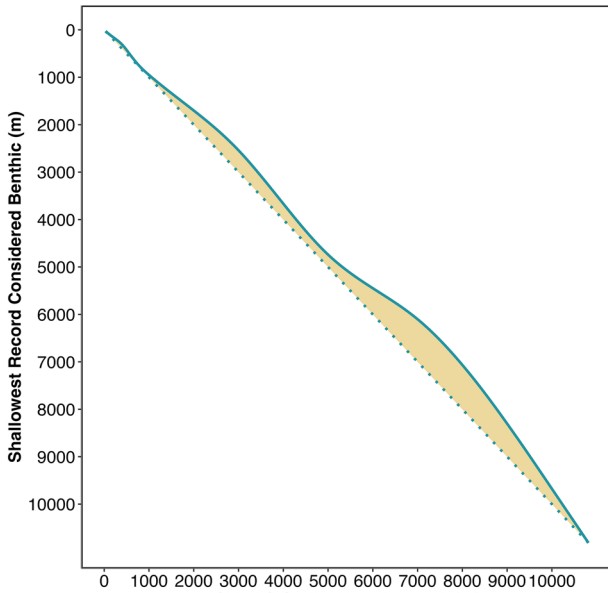

**Fig. 1 | Visualization of the benthic data cloud across the bathymetric gradient.** The dashed line represents GEBCO seabed depth for any given location; the solid line represents the shallowest point at which an observation is considered benthic (the benthic minima) for that seabed depth, with the shaded area in between identifying the benthic data cloud, the area within which points were classed as benthic.

In the pursuit of developing a comprehensive dataset, a vital step is gaining a profound understanding of existing information. As defined by ref. 15, ocean biodiversity informatics "*is the use of computer technologies to manage marine biodiversity information, including data capture, storage, search, retrieval, visualisation, mapping, modelling, analysis and publication*". The Ocean Biodiversity Information System (OBIS[16]; www.obis.org) is the most extensive repository of data from depths below 30 metres with approximately 19 million records, surpassing the Global Biodiversity Information Facility's (GBIF; www.gbif.org) collection of around 16 million records. Despite underrepresentation of the deep sea in OBIS[17], the applications of the OBIS dataset are manifold, ranging from scrutinizing species distribution to delineating biogeographies and informing management decisions (e.g. refs. 18–20.). However, a pitfall of both the OBIS and GBIF datasets is the absence of an automated method for segregating benthic and pelagic data; a limitation traditionally addressed through manual sorting when working with subsets of the data[21]. A reasonable proposition to overcome this problem may be to only extract records for benthic groups e.g., stony corals (Scleractinia). However, the presence of planktonic records in both databases means distribution maps created from these data are often not representative of where adult life stages are found. This lack of ability to distinguish benthic from pelagic 'big data' is problematic given the distinct responses of different groups to environmental changes, where the comparative horizontal and vertical mobility of pelagic species is a key consideration[22,23]. This distinction is particularly important because pelagic species, with greater horizontal mobility, may be able to track shifting environmental conditions more readily than benthic organisms, which are often sessile or have limited vertical and horizontal dispersal capacity. As a result, their vulnerability and ecological responses to threats can differ markedly.

In this study, we address and rectify this gap, providing an automated solution to benthic-pelagic classification of open-access biological data. While global data aggregators such as OBIS and GBIF already support a wide range of fundamental research, our curated global dataset offers a pathway to improve the habitat-specific accuracy of these records, helping to better answer fundamental questions about life in the deep ocean.

## Results

### Developing an automated solution

Segregation of benthic and pelagic data was accomplished through characterising the relationship between the record depth as *per* its entry in OBIS, and the seabed depth at the same location derived from the General Bathymetric Chart of the Ocean (GEBCO[24]). Although the deep sea is commonly defined as those areas deeper than 200 m, the challenges associated with collecting data deeper than standard SCUBA depths are similar (and typically increase with depth), and therefore, for the purposes of analysis, we include all records from 30 m and below. For the 18.9 M records sitting atop the GEBCO bathymetric grid and with OBIS record depths between 30 and 10,920 m, we created a subset for model training. To do this, we selected records whose GEBCO-derived depth matched one of 14 predefined depth horizons (i.e., GEBCO depth is 100 m, 1000 m etc.). These discrete depth layers were chosen to ensure even coverage across the bathymetric range and reduce potential biases in depth distribution during model building. For each subset, partitioning around medoids (PAM) clustering on points outside of the 95th percentile identified the benthic data cloud, resulting in a minimum record depth considered to be the threshold at which data can be classed as pelagic. Generalized additive modelling (GAM) was used to mathematically characterise the relationship between the 14 benthic minima values and the GEBCO depth of their subsets (Supplementary Notes 1), allowing extrapolation of the relationship across the full 18.9 M records to identify the benthic data cloud across the bathymetric gradient (Fig. 1). The resultant benthic and pelagic datasets comprising 12.7 and 6.2 million records respectively, represent the most comprehensive classification of marine biodiversity data to date.

### Patterns in the data

Filtering the global OBIS dataset into its constituent parts revealed key differences between available pelagic *versus* benthic data (Fig. 2). It is clear that the spatial coverage in OBIS largely represents spatial variability in pelagic sampling, despite higher numbers of benthic records.

Benthic record density is highest around western Europe, the United States east coast and New Zealand (Fig. 2B). Whilst western Europe and to a lesser extent the US east coast remain hotspots for pelagic data, high numbers of records are also identified from the northern Gulf of Mexico (Fig. 2C). Almost all sections of coastline have undergone some level of benthic and pelagic sampling according this these data, although much of these areas host ≤100 records per 1° grid cell and thus our ability to draw meaningful scientific conclusions is limited.

Further offshore in areas beyond national jurisdiction (ABNJ; more than 200 nm from land), benthic records in OBIS become comparatively extremely sparse. Benthic records here are mostly associated with notable topographic features such as ridges and seamount/seamount chains including the Louisville Ridge to the east of New Zealand, the Southwest Indian Ridge to the south of Madagascar and the Salas Y Gomez and Nazca ridge systems to the west of Chile, but these are present in relatively low densities (≤100 records per 1° grid cell). Excluding areas on/around these features, bathyal, abyssal and hadal benthic environments of the ocean in ABNJ are particularly undersampled. Here, areas with benthic data, albeit low-density, include the Hatton-Rockall Basin and the abyssal region of the southeast of the Chagos Archipelago, with the remaining space mostly baron of open-access records. This is somewhat contrasting to the spatial distribution of pelagic data in ABNJ. Here, it appears that most grid cells in the northern hemisphere host at least one pelagic record, with 'well-sampled' areas (again, with low record densities) being located off the eastern US, the Arabian Sea, and the northwest Pacific.

Stratification of both the benthic and pelagic datasets across latitude and depth, the key variables across which an idealised global dataset would be stratified, revealed large biases (Fig. 3). Over 79% of benthic data from 30 m and deeper in OBIS comes from the northern hemisphere. In the global south, record density peaks between ~40° and 50° (Fig. 3), representing comparatively high record numbers around New Zealand and on

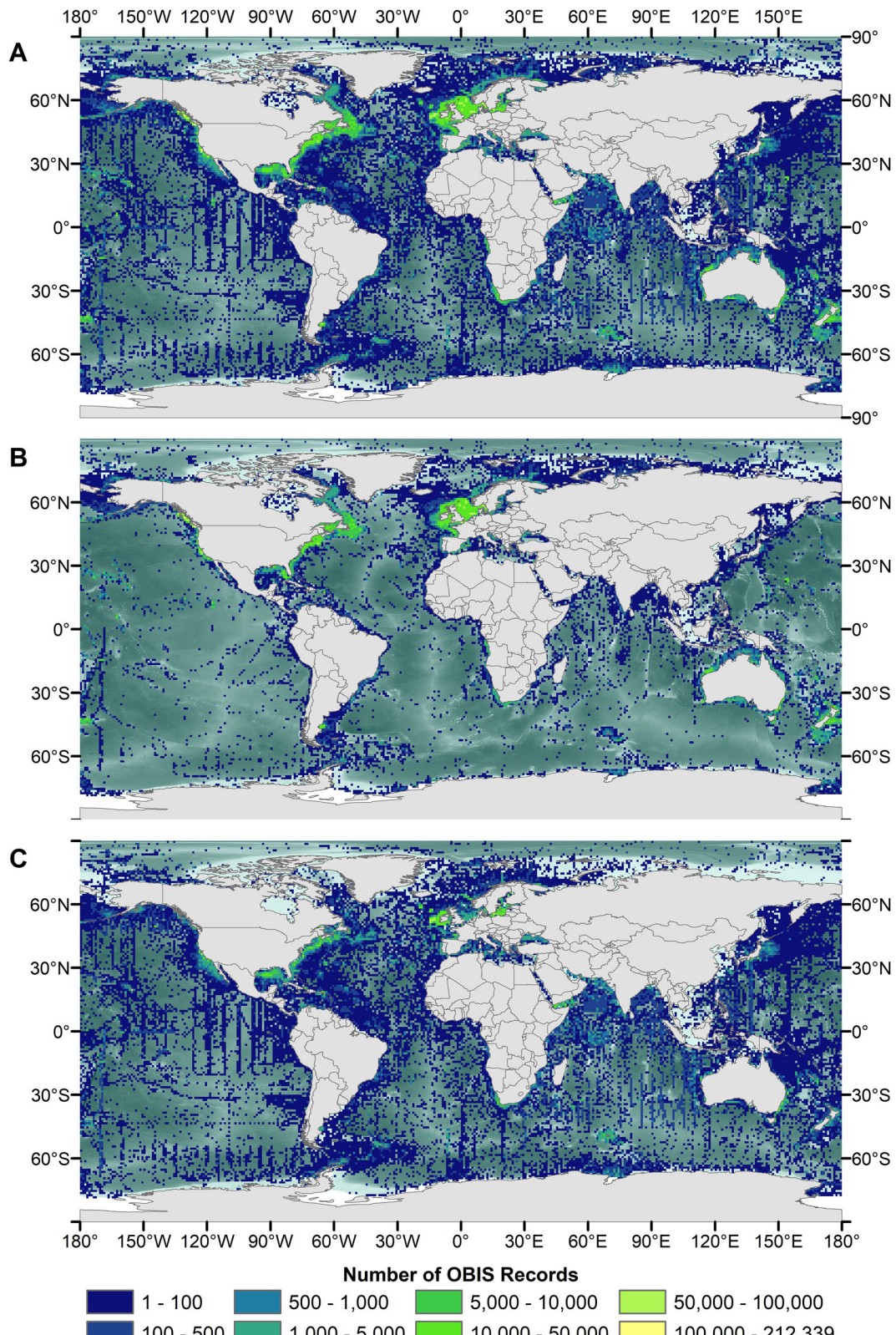

**Fig. 2 | Mapped global Ocean Biodiversity Information Service data for Animalia.** Data are gridded by number of records per 1 degree x 1 degree cell: **A** – benthic and pelagic data combined ($n$ = ~18.9 M), **B** – benthic data ($n$ = ~12.7 M), **C** – pelagic data ($n$ = ~6.2 M).

the Patagonian Shelf (Fig. 2B). Similarly, for pelagic data, over 76% derives from the northern hemisphere.

Across the bathymetric gradient, an almost exponential decrease in the number of benthic and pelagic records is observed, with the median OBIS record depth of the benthic dataset being 107 m (Fig. 3). Thus, 50% of the

benthic data in OBIS come from the shallowest 0.98% of the seafloor. Again, the case is much the same for pelagic data with a median of 80 m, and 98.9% of pelagic records from OBIS are from the upper 2000 m of water column.

Distribution of records across major phyla reveal a distinct bias towards Chordata in both the pelagic and benthic datasets, perhaps

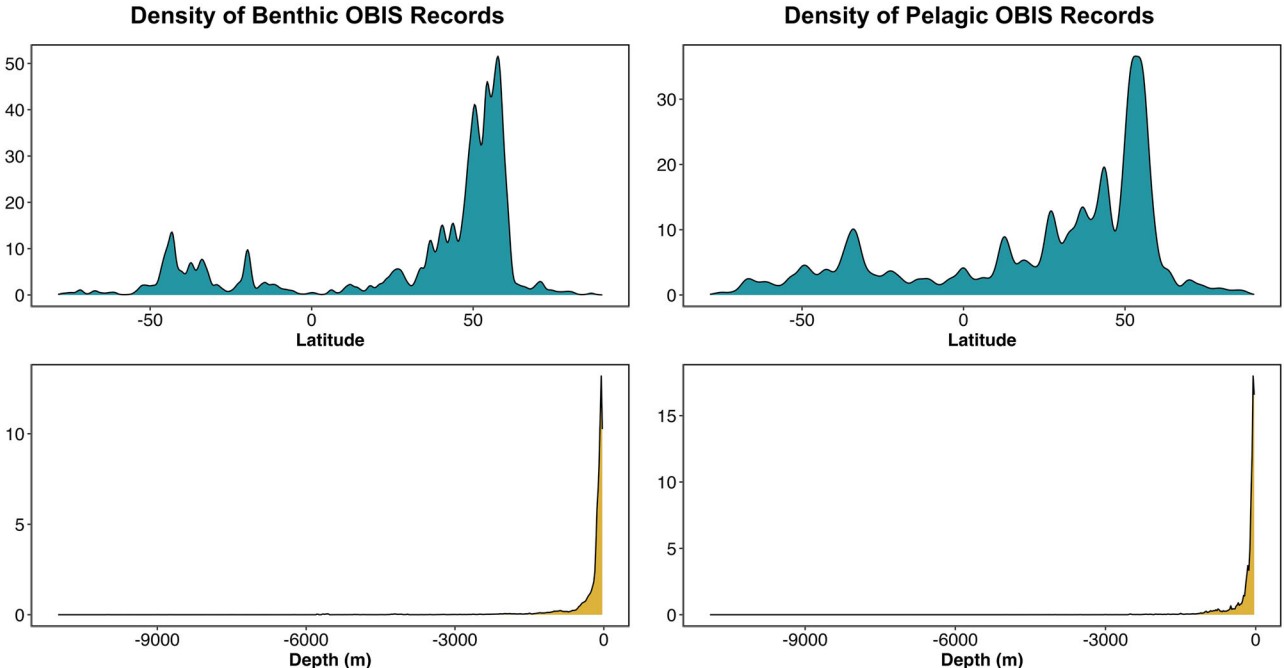

**Fig. 3 | Density of global Ocean Biodiversity Information Service records against latitude and depth.** Kernel density values were multiplied by 1000 for ease of reading.

unsurprising given the popularity of trawl sampling as a sampling methodology that naturally targets fish species (Figure 4). Whilst Arthropoda is the second most abundant phylum represented in both datasets, they make up a far larger proportion of the pelagic dataset than they do the benthic. For the latter, Arthropoda is the only invertebrate phylum with >1 M records globally (1.33 M) – all other invertebrate phyla contain less than 1 M benthic records. As many records are not identified to species level, it is not possible to accurately assess how well different phyla are represented in terms of species diversity.

## Discussion

Data for offshore (>30 m) ecosystems in open-access global repositories are heavily biased towards the northern hemisphere, shallow depths, and vertebrates. Mapping benthic and pelagic data separately highlights how spatial coverage statistics and patterns derived from the full dataset are heavily influenced by pelagic data, despite being numerically dominated by benthic records. Whilst our findings lend support to [16]'s suggestion that midwater habitats are under-recorded relative to surface waters and the seabed in terms of the number of records, our data show that this is not true for spatial coverage, with the geographic distribution of benthic records being significantly more constrained than that of the pelagic data (Fig. 2B *versus* Fig. 2C).

There are three potential explanations for the observed decrease in openly-accessible data with depth: (1) the deep ocean is low in standing-stock, and thus low record numbers reflect this as opposed to a lack of sampling; (2) the majority of deeper data are not in OBIS and hence the patterns here do not reflect the actual sampling effort; and (3) these environments are truly under-sampled.

The availability of the data presented here could be explained by a combination of all three. Whilst the deep sea was considered baron by some early naturalists (Forbes' azoic hypothesis[25]), studies from the 1960s began to dismantle the idea of a 'low diversity and biomass deep sea'[26,27]. This said, there is a well-established decrease in animal densities and biomass with depth for both benthic[28] and pelagic taxa[29] that could contribute to our observations. As there is no standardised global product for sampling effort, it is difficult to robustly assess how variation in effort influences observed patterns, or the degree to which non-submission of data to platforms like OBIS may contribute to spatial biases. While recent studies have demonstrated that proxies for effort, such as the number of unique sampling events

over time, can be derived from OBIS (e.g.,[30]), such analyses are beyond the scope of the current study. Nevertheless, discrepancies between mapping open-access biodiversity data and literature-derived sampling effort have been noted for some large ocean areas, although overarching spatial patterns are often broadly consistent[31,32]. Whilst explanations one (decline in animal standing stock with depth) and two (missing data in OBIS) likely *influence* the dataset, we do not feel they are solely responsible for the patterns in the data. Therefore, we conclude that explanation three – that these environments are truly under-sampled – likely has a greater influence on the observed patterns than the other explanations.

It is clear from visual assessment of benthic and pelagic datasets that we have not fully succeeded in completely separating benthic from pelagic data using this method. In particular, the presence of continuous 'lines' of data points, such as those visible in the Indian Ocean directly south of the Arabian Sea (Fig. 2B), suggests that some pelagic records remain within the benthic dataset. These linear patterns are characteristic of data collected using towed sampling gear such as plankton trawls or midwater nets, which are commonly deployed over long distances in pelagic surveys. In contrast, benthic sampling, especially in the deep sea, typically involves discrete sampling events at specific locations (e.g., grabs, cores, ROV deployments) rather than continuous transects. Therefore, the spatial structure of these data strongly indicates a pelagic origin. Query of example data points confirms this suspicion but suggests these data have simply been entered incorrectly into OBIS with the depth at the seafloor at the point of sampling entered into OBIS, rather than the depth at which the sample was taken. Manual cleaning of these data could be undertaken to improve data quality. However, automation of further cleaning of such erroneous data is more difficult but could potentially employ more advanced models, including artificial intelligence, to look at sample point behaviour, in this case data 'lines', and assign these points as pelagic in a further data pipeline step.

Despite the inevitable errors in such as vast database, open-access global data repositories like OBIS are undeniably useful but should be used with caution, not least of all because they aggregate data collected using a wide range of methods, sampling designs, and research objectives. Although the 18.9 M records analysed here split into approximately two thirds benthic and one third pelagic, the spatial variability in each data type is great (Fig. 2). Products resulting from the analysis of an unfiltered OBIS dataset, for example biogeographic classifications, could be

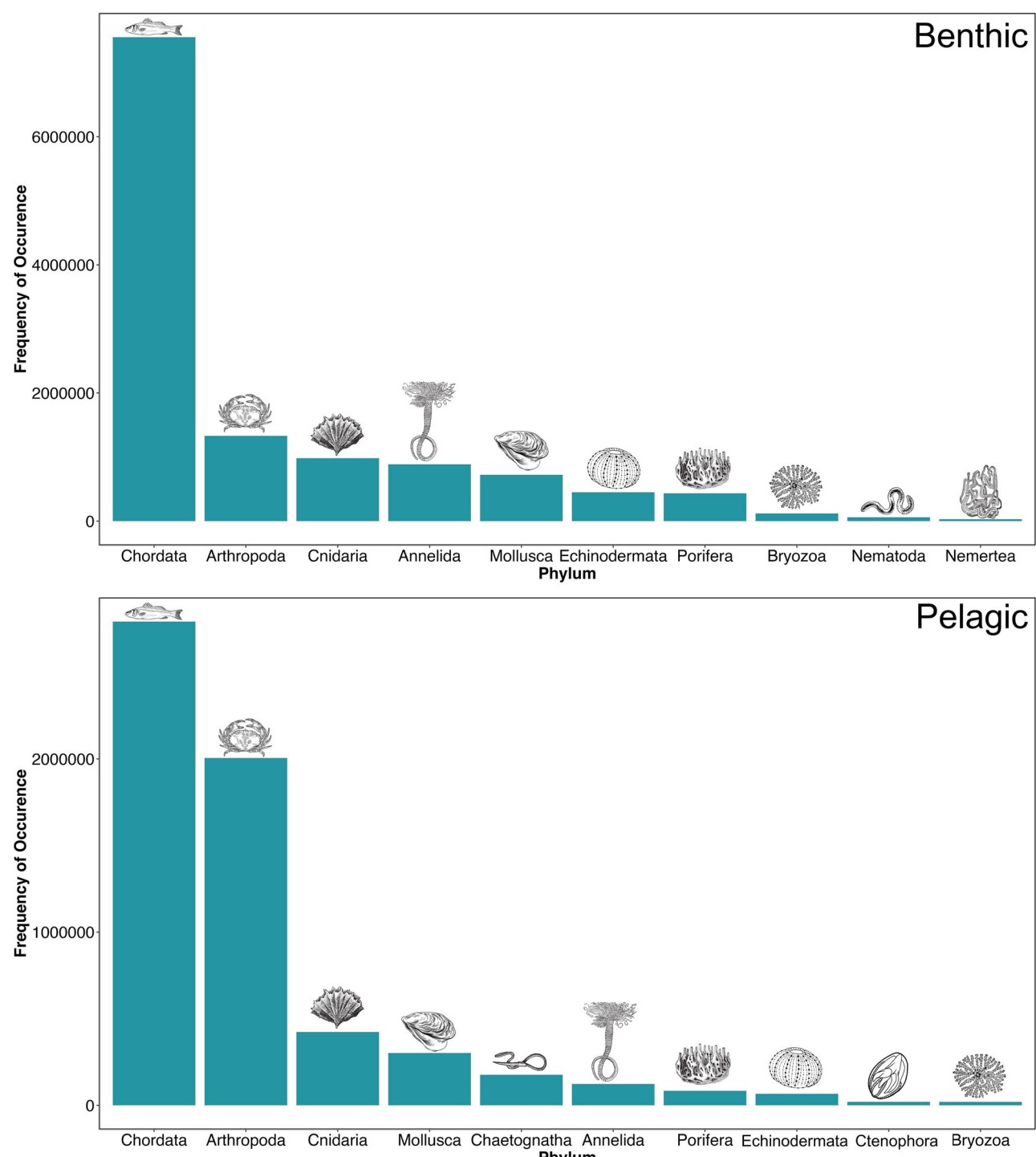

**Fig. 4 | Top ten most abundant phyla in the benthic and pelagic global datasets.** Illustrations of each group are shown above each bar. Note the different y axis values given the different dataset sizes.

misleading given ecological boundaries and their drivers can be vastly different for benthic and pelagic taxa (e.g. ref. 33 compared to ref. 34). This is in addition to the known spatial biases in the databases themselves sometimes influencing the data products for which they are used[35]. Therefore, whilst the accessibility of open-access repositories is to be celebrated, this work demonstrates that to further realise the potential of ocean biodiversity informatics, we should: (1) work to encourage owners to deposit their data in open-access repositories; and (2) utilise data manipulation pipelines, such as the one proposed here, that allow statistically-evidenced sorting of benthic and pelagic records.

The consequences of biases in open-access global data towards the northern hemisphere, shallow waters and vertebrates mean any modelled inferences and predictions made using these are likely over-fitted to more heavily sampled areas/taxa[35]. [4] propose latitude and depth as two climate-related variables against which a comprehensive deep-ocean dataset should be stratified to allow quantification of biodiversity and characterisation of ecological niches. The critical lack of accessible data, combined with the unequal stratification of what little data *is* available across both variables (Fig. 3), mean that we are currently unequipped to answer fundamental questions that remain regarding life's distributional drivers across most of

the planet's surface. These knowledge gaps hinder our ability to establish effective baselines, in turn impeding our ability to forecast how diversity and the services it provides (e.g., fisheries, climate regulation etc.) might change based on increasing anthropogenic pressures[2]. Under sampling of the southern hemisphere is particularly concerning, given that highly impactful pressures here are fast-increasing, rendering ecosystems here the most-threatened but least studied[36].

Based on our results, future exploration efforts should target the following: (1) ABNJ, (2) depths below 1,500 m, (3) the southern hemisphere, and (4) invertebrates. Collection of data from these areas would, at minimum, lessen some of the biases within the existing data and move us closer to robust quantification of biodiversity and characterisation of ecological niches globally. Prioritisation of the tropics and global south in sampling efforts will help alleviate the 'most-threatened least studied' paradigm; however, it is critical that such regimes be co-designed and delivered by scientific communities from across the world, including from under-represented groups from the tropics and global south[37,38]. Failure to do this will only promote further biases in science, ultimately jeopardising equitable and effective management[39].

The UN Decade of Ocean Science for Sustainable Development (A/RES/72/73) calls on us to undertake '*the science we need for the ocean we want*'. This necessitates that we fill the data gaps for offshore and deep-ocean ecosystems, a cause championed by the Challenger 150 UN Ocean Decade Programme. Doing so would provide much-needed evidence to underpin numerous international conservation efforts, including decisions surrounding the placement of MPAs in ABNJ under the High Seas Treaty. Without data on what species and habitats are present, we cannot truly gauge how representative or connected MPA networks are/will be, thus hindering our achievement of the $30 \times 30$ initiative, the Kunming-Montreal Global Biodiversity Framework target to effectively conserve and manage at least 30% of marine areas. Despite global efforts since their signing in 2011, none of the 20 Aichi Biodiversity Targets were fully achieved by the 2020 deadline[40]. To achieve subsequent global targets, strategic prioritisation of data collection is needed. The dataset reported in this manuscript should serve as a starting point from which to make decisions, drive future sampling strategies and ultimately support sustainable management.

## Methods

### Downloading & subsetting the data

Downloading and filtering methods proposed in this study comprise 10 steps (Supplementary Fig. 1). Firstly, records for 'Animalia' and its taxonomic children were downloaded using the occurrence() function from the 'robis' package[41] in R[42] between June 27th and June 30th 2023; the 'startdepth' argument was employed to extract records at 30 m water depth and below. This initial download protocol extracted 19.1 M records from the OBIS database.

Using the extract() function from the 'raster' package[43], depth values from the General Bathymetric Chart of the Oceans (GEBCO) 2020 gridded bathymetry[24] were extracted for all OBIS record locations. Records with no GEBCO depth, typically those on land with the location registered as a museum, were removed from analysis, resulting in 18.9 M of the initial 19.1 M downloaded records being used in further analyses.

To distinguish between benthic and non-benthic records, the relationship between original record depth and GEBCO depth had to be characterised across the bathymetric gradient. Records were subsetted for specific depth horizons/bands where the GEBCO depth was a set value or small range (Supplementary Table 1). Much of the variability in each subset was consumed with data past the 95th percentile – these points were often erroneous records with depths far greater than the bathymetry suggested, and therefore these data were removed.

### Clustering data

Cluster analysis was employed to identify groups of points within each subset, with the group of points containing the GEBCO depth value assumed to represent the benthic data point cloud. This approach was adopted given sampling methodologies typically either target the benthos or remain firmly within the pelagic; for example, there are trawls targeting the mid-water, or bottom. There are very few methodologies whereby, in water depths greater than 200 m, non-benthic sampling methodologies and equipment would be reaching within ~100 m of the seafloor, given the safety implications of snagging. Therefore, it is reasonable to assume that the data structure might reflect this, and thus clustering is a suitable technique to extract benthic data points.

Prior to the cluster analysis, all individual variables were normalised between 0 and 1. An unsupervised, non-hierarchical clustering algorithm (Clustering Large Applications, CLARA) capable of working with large datasets was employed for the analysis via the pamk() function in the 'fpc' package[44].

Cluster analyses were carried out, iteratively testing 2 to 10 clusters for each depth horizon subset. Average silhouette width (ASW, 0-1) provides an indication of the similarity of an object in relation to its own cluster, compared to other clusters. ASW metrics for each subset were used to statistically identify optimal clustering solutions (Supplementary Fig. 2). Cluster solutions assigning benthic or pelagic status to data points were plotted for all iterations for each subset (examples in Supplementary Fig. 3). The final clustering solutions and resulting benthic maximum and minimum values (Supplementary Table 2) were selected based on ASW scores, assessment of the resulting plots and expert opinion.

### Modelling the benthic data cloud

The upper limit of the cluster containing the GEBCO depth for which the data were subsetted for was adopted as the upper threshold of what was considered benthic data at that depth horizon; anything above which was removed on the assumption it represented pelagic data. The benthic minimum thresholds for the 14 set depth horizons were used to build a generalised additive model (GAM), build using the 'mgcv' package[45] (Supplementary Notes 1). A GAM was selected as the model of choice given its flexibility in capturing complex relationships in ecological data. The response variable, benthic minimum, was modelled using a Gaussian family and an identity link function. The model included a single smooth term for GEBCO depth and explained 100% of the deviance in the training data. The smooth term was highly significant (edf = 8.27, F = 98,627, $p < 0.01$), indicating a strong, non-linear relationship (Supplementary Table 3). Model diagnostics (posterior predictive check, residual plots) performed using the check_model() function from the 'performance' package[46] suggested a good overall fit, though some heteroscedasticity was noted at low fitted values, consistent with expectations for ecological GAMs. Given the derived nature of the response variable, the high deviance explained is not unexpected. To further understand model performance, the 95% confidence intervals at each of the 14 predetermined depth horizons were checked (Supplementary Table 4).

The model elucidated mathematical characterisation of the upper limit of the benthic data cloud across the depth gradient and allowed that relationship to be predicted across the full 18.9 M extracted OBIS records to extract benthic data (Supplementary Table 5).

### Spatial analysis

For mapping purposes, data were binned into 1° x 1° grid cells and the number of OBIS records was summed per cell. Stratification of data against depth and latitude was done so using kernel density values.

### Data availability

The final benthic and pelagic datasets are available to download on Zenodo: https://zenodo.org/records/15487410. The individual datasets submitted to OBIS that contribute to the final datasets presented here can be found at https://doi.org/10.25607/obis.export.55a1b2d3.

### Code availability

Code required to repeat the analysis on the global OBIS dataset is available in a public GitHub repository: www.github.com/ameliabridges/Bridges-Howell_OA_repo_pipeline.

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

## Acknowledgements
The authors would like to thank Dr Nils Piechaud for contributing to preliminary work with the OBIS database. This work was funded by (1) European Union's Horizon 2020 'Mission Atlantic Project' No. 862428 (KH), and (2) United Kingdom Research and Innovation Global Challenges Research Fund 'One Ocean Hub Project' NE/S008950/1 (KH, AB).

## Author contributions
Conceptualization: A.E.H.B., K.L.H. Methodology: A.E.H.B., K.L.H. Formal analysis: A.E.H.B. Writing – original draft: A.E.H.B. Writing – review & editing: A.E.H.B., K.L.H. Funding acquisition: K.L.H.

## Competing interests
The authors declare no competing interests.
