## [Transparent Peer Review file · Communications Earth & Environment]

Prioritisation of ocean biodiversity data collection to deliver a sustainable ocean

Corresponding Author: Dr Amelia Bridges

Version 0:

Decision Letter:

Dear Dr Bridges,

Your manuscript titled "Prioritisation of ocean biodiversity data collection to deliver a sustainable ocean" has now been seen by 2 reviewers, and we include their comments at the end of this message. They find your work of interest, but some important points are raised. We are interested in the possibility of publishing your study in Communications Earth & Environment, but would like to consider your responses to these concerns and assess a revised manuscript before we make a final decision on publication.

We therefore invite you to revise and resubmit your manuscript, along with a point-by-point response that takes into account the points raised. Please highlight all changes in the manuscript text file.

Please submit your point-by-point responses as a separate file, distinct from your cover letter where you can add responses to the Editors' comments that you do not want to be made available to the reviewers. Word files are preferred. We recommend that any figures, tables or graphs that are included in the response to reviewers are also included in the main article or Supplementary Information.

Please use the following link to submit your revised manuscript, point-by-point response to the referees' comments (which should be in a separate document to any cover letter), a tracked-changes version of the manuscript (as a PDF file) and the completed checklist:

Link Redacted

We hope to receive your revised paper within six weeks; please let us know if you aren't able to submit it within this time so that we can discuss how best to proceed. If we don't hear from you, and the revision process takes significantly longer, we may close your file. In this event, we will still be happy to reconsider your paper at a later date, as long as nothing similar has been accepted for publication at Communications Earth & Environment or published elsewhere in the meantime.

Please do not hesitate to contact us if you have any questions or would like to discuss these revisions further. We look forward to seeing the revised manuscript and thank you for the opportunity to review your work.

Best regards,

Alice Drinkwater, PhD
Associate Editor
Communications Earth & Environment

EDITORIAL POLICIES AND FORMATTING

Editorial Policy: [Policy requirements](https://www.nature.com/documents/nr-editorial-policy-checklist.pdf) (Download the link to your computer as a PDF.)

- Behavioural and social science
- Ecological, evolutionary & environmental sciences
- Life sciences

<https://www.nature.com/documents/nr-reporting-summary.zip>

Furthermore, please align your manuscript with our format requirements, which are summarized on the following checklist: [Communications Earth & Environment formatting checklist](https://www.nature.com/documents/commsj-phys-style-formatting-checklist-article.pdf)

and also in our style and formatting guide [Communications Earth & Environment formatting guide](https://www.nature.com/documents/commsj-phys-style-formatting-guide-accept.pdf) .

***** DATA:** Communications Earth & Environment endorses the principles of the Enabling FAIR data project (<http://www.copdess.org/enabling-fair-data-project/>). We ask authors to make the data that support their conclusions available in permanent, publically accessible data repositories. (Please contact the editor if you are unable to make your data available).

All Communications Earth & Environment manuscripts must include a section titled "Data Availability" at the end of the Methods section or main text (if no Methods). More information on this policy, is available at <http://www.nature.com/authors/policies/data/data-availability-statements-data-citations.pdf>.

If a community resource is unavailable, data can be submitted to generalist repositories such as [figshare](https://figshare.com/) or [Dryad Digital Repository](http://datadryad.org/). Please provide a unique identifier for the data (for example a DOI or a permanent URL) in the data availability statement, if possible. If the repository does not provide identifiers, we encourage authors to supply the search terms that will return the data. For data that have been obtained from publically available sources, please provide a URL and the specific data product name in the data availability statement. Data with a DOI should be further cited in the methods reference section.

REVIEWER COMMENTS:

Reviewer #2 (Remarks to the Author):

Overview: This paper presents an important contribution, by developing an automated pipeline to classify records from the Ocean Biodiversity Information System as benthic or pelagic. Such information is critical to many works, and here the authors used that information to assess spatial (across depth and latitude) and taxonomic (phylum) bias in this global database. They show that there are strong bias towards shallow areas and the northern hemisphere.

Although this work is significant, I believe there are some points that need to be clarified/better detailed before the paper can be published. I also think that the methods, especially the model fitting, need to be expanded and further detailed.

One critical, which does not impede the publication of this piece, is that the authors make no (or do not present) a sensitivity analyses based on taxon. Many species are exclusively benthic or pelagic. Authors could have selected a subset of the data to test to what extent all records of a certain species were being correctly classified as benthic or pelagic. The authors note that there are records also from planktonic/larval stages which does not reflect the adult stage of the organism. That is true, but dataset information can be used to assess which type of data the dataset was targeting. Again, this could be done for a subset of the data to provide a further assessment of the capacity of classification of this model.

One further point that I would like to highlight is that the authors say that they are providing a dataset as one of the products of the work. This is great, and those curated datasets are always good for the community. But more important, and what I believe they should highlight more, is the pipeline. With over a million records being added monthly to OBIS, a new dataset will be needed soon. Thus, the real value is in the pipeline which will be further used in the future.

Title: could be changed to reflect more the core of the work: the method and the bias found. But just a suggestion.

Line 48: please, provide examples of which key drivers you are referring here.

Line 49: The wording “biologically irrelevant” may be wrong here. Think, for example, on pressure differences created by depth and how this affects life.

Line 74-75: that is true, but it should be noted that records, in general, contains extensive additional information attached to it that could also be used to disentangle that.

Line 81-83: this needs rephrasing; OBIS and GBIF are global data aggregators, unifying multiple data collection efforts, including a wide range of methods and initial research objectives. Both are already serving to answer fundamental questions, including on deep sea. But, you are now clearly showing that we have a pathway to improve data for the deep sea, what is extremely important.

Line 81: maybe classification instead of filtering?

Lines 89-91: this could be more clearly explained here. Reading the rest of the methods I came to understand that, but on the first reading no.

Line 94: add the acronym (GAM)

Line 99: not sure if we can call this a “deep ocean” dataset; maybe it would be better to say that represent the most comprehensive classification of marine biodiversity data.

Line 116: “this these” – revise.

Line 157-160: the authors could expand this part by also highlighting the numbers of known species in each of those groups. Looking at WoRMS (<https://www.marinespecies.org/aphia.php?p=stats>), there are ~24,000 marine Chordata, while only Mollusca is > 85,000 and Arthropoda > 73,503. It would be nice to have an overview of how taxonomically biased the pelagic/benthic datasets are (here you talk only about # of records, what about the number of species represented?).

Line 167: bias -> biased

Line 175-179: the authors propose good causes here, well chosen.

Line 187-188: indeed, there is no global product of sampling effort, but it is possible to derive metrics from OBIS in that regards. For example one can easily calculate the sampling effort as the number of unique events in a single day or month. See for example <https://doi.org/10.1111/ecog.07006> (Thyrring et al. 2024). It would be interesting to derive such a metric for at least a sample of the grid cells you are analyzing, as this would enrich your discussion. If you can't do that assessment now, or believe is out of scope, then I advise to rephrase that part, highlighting that effort could be at play here.

Lines 181-194: this whole paragraph is confusing. First the authors says that “the data presented here could be explained by a combination of all three of these possible explanations”. Then they say that “we feel it is unlikely that the decline in animal standing stock with depth adequately explains our observations, or that incorporating data that is absent from OBIS into our spatial analysis would significantly change our findings. We therefore conclude these environments are truly under-sampled.”. I agree that the three must be at play together in shaping the data, but the final phrases are not reflecting that.

Lines 196-206: please, provide a more detailed explanation for the reader of what you mean with the “lines” and why this would be connected to pelagic sampling.

Lines 208-209: again, I think the authors should better explain to the user why they should have caution. That is not a

problem, but the nature of OBIS, GBIF: they are global aggregators of data, collected through multiple types of methods and across many different objectives. As so, any research done with that must be done with the appropriate care. Data must be filtered and considered according to the use case.

Lines 222-224: over-fitted is not a good term here. Over-fitted refers always to model fitting “too much” to the input data, thus lacking the capacity to predict to new data. Here better to say “biased towards”

Lines 232-234: indeed, this is critical and an important find of this work!

Methods

Lines 429-432: The justification for using 30m is acceptable, but I would not say the challenges of collecting data at 100m depth are the same as collecting at 1000m. This definition should also be on the main text, so readers now why <30m is being considered deep-sea by you.

Lines 442-443: I could not understand what you meant here until I went to the ED. Needs rephrasing and more detail.

General: when the target is prediction, it is generally advisable to have some sort of cross-validation (including leave-one-out). Also, a model with 100% of deviance explained appears to be over-fitted and without a validation in an external dataset it is impossible to assess if it is correct or not (I'm definitely not saying that your model is wrong because of the 100% deviance explained). I think the authors need to provide more details about the model fitting and how model was assessed.

Reviewer #3 (Remarks to the Author):

Review of COMSENV-24-4052-T “Prioritisation of ocean biodiversity data collection to deliver a sustainable ocean”
General comment:

The authors provide an automated data pipeline for extracting and separating benthic from pelagic data in open-access repositories. The author's results showed that large spatial and taxonomic biases exist in both the benthic and pelagic global datasets, favouring depths shallower than ~100 m, the northern hemisphere, and vertebrate species. The authors suggested a newly compiled, cleaned, and classified dataset to be used to identify areas of chronic under sampling and high priority regions for exploration. They recommended a coordinated strategic prioritisation of sampling as needed approach to support modelling and prediction, better management and understanding of marine life.

The introduction, methodology, results, discussion, and conclusion are generally sufficiently presented. Indeed, the manuscript needs major changes before being accepted. Details are listed below:

Abstract:

Lines 6, 11: please check the author guidelines, usually reference are not allowed in the abstract as it should be a self-contained summary of author's research

Introduction

Lines 22-23: we had a huge advances in technologies and exploration in the last decades, and it is probably to extreme this sentence without any recent references. Please rephrase and/or add recent references

Lines 36-41: the role of data is already clear, maybe authors wanted to convey a different message. I would suggest to clarify and add details if necessary.

Lines 48-49: please develop more the sentence explaining the reason why you consider “latitude and depth biologically irrelevant”

Lines 52-53: please correct: temperature typically decrease with increasing depth

Lines 53-55: the sentence seem to be repeated. Please rephrase it correctly.

Lines 78-80: please remove or move to introduction, and add the correct information. These are no information related to Data sources and interpretation of the indicators, as indicated in the subtitle.

Lines 55-58: not very clear the message the authors want to convey. Please clarify

Line 64: please correct the web address

Lines 64-69: OBIS and GBIF are interconnected in the last years, so I believe that it is not very useful the comparison anymore. I would suggest removing or rephrasing it

Line 77-78: I would suggest developing more why the comparison horizontal vs vertical is a key consideration.

Materials and Methods

Lines 86-99: please add the level of uncertainty of the automated solution.

Figure 1: please increase font size, not readable

Results

Lines 109-132: the variability could be strongly due to users, which decide what and where submit the data. Please, consider this aspect in the section

Figure 2: please increase font size, not readable. In addition consider to use colour blind friendly palette

Line 156: please develop more the last part of the sentence otherwise remove it since it not scientifically demonstrate. In addition, “popularity”? I would suggest using a different term

Figure 3-4: please increase font size, not readable

Discussion

Lines 171-173: please develop more the reason why your data are in contrast to Webb' suggestion

Lines 191-194 vs 181.182: please read again and correct

Lines 199-201: due to incorrect logging or OBIS submission form

Communications Earth & Environment is committed to improving transparency in authorship. As part of our efforts in this direction, we are now requesting that all authors identified as 'corresponding author' create and link their Open Researcher and Contributor Identifier (ORCID) with their account on the Manuscript Tracking System prior to acceptance. ORCID helps the scientific community achieve unambiguous attribution of all scholarly contributions. You can create and link your ORCID from the home page of the Manuscript Tracking System by clicking on 'Modify my Springer Nature account' and following the instructions in the link below. Please also inform all co-authors that they can add their ORCIDs to their accounts and that they must do so prior to acceptance.

Version 1:

Decision Letter:

Dear Dr Bridges,

Your manuscript titled "Prioritisation of ocean biodiversity data collection to deliver a sustainable ocean" has now been seen by our reviewers, whose comments appear below. In light of their advice we are delighted to say that we are happy, in principle, to publish a suitably revised version in Communications Earth & Environment.

We therefore invite you to revise your paper one last time to address the remaining concerns of our reviewers. At the same time we ask that you edit your manuscript to comply with our format requirements and to maximise the accessibility and therefore the impact of your work.

EDITORIAL REQUESTS:

****Please take care to match our formatting and policy requirements. We will check revised manuscript and return manuscripts that do not comply. Such requests will lead to delays. ****

SUBMISSION INFORMATION:

OPEN ACCESS:

Communications Earth & Environment is a fully open access journal. Articles are made freely accessible on publication. For further information about article processing charges, open access funding, and advice and support from Nature Research, please visit <https://www.nature.com/commsenv/open-access>

At acceptance, you will be provided with instructions for completing the open access licence agreement on behalf of all authors. This grants us the necessary permissions to publish your paper. Additionally, you will be asked to declare that all required third party permissions have been obtained, and to provide billing information in order to pay the article-processing

charge (APC).

Link Redacted

Best regards,

Alice Drinkwater, PhD
Associate Editor
Communications Earth & Environment
Consulting Editor
Communications Sustainability

REVIEWERS' COMMENTS:

Reviewer #2 (Remarks to the Author):

I appreciate the authors' effort to address all the comments. As I noted in my initial review, this paper makes an important contribution, and I now believe it is ready for publication.

I just highlight one point that I think could still be adjusted (very minor):

- In the paragraph 233-259, the changes made were really good. I just feel that the last sentence is sounding strange. "holds true", in my opinion, brings the idea of confirmation, while all three are potential explanations to the decrease in data availability ("There are three potential explanations..."). Now, your results point out that the third hypothesis (under sampling) is probably contribution more to the problem. Something like that, maybe: "Therefore, we conclude that explanation three — that these environments are truly under-sampled — likely has a greater influence on the observed patterns than the other explanations."

Reviewer #3 (Remarks to the Author):

Thanks to the coauthors for having considered the comments/suggestions and improved the manuscript. I have no further relevant comment to submit.

Manuscript title: Prioritisation of ocean biodiversity data collection to deliver a sustainable ocean

Reviewer #2		
Section	Reviewer's comment	Author's response
Remarks to authors	Overview: This paper presents an important contribution, by developing an automated pipeline to classify records from the Ocean Biodiversity Information System as benthic or pelagic. Such information is critical to many works, and here the authors used that information to assess spatial (across depth and latitude) and taxonomic (phylum) bias in this global database. They show that there are strong bias towards shallow areas and the northern hemisphere. Although this work is significant, I believe there are some points that need to be clarified/better detailed before the paper can be published. I also think that the methods, especially the model fitting, need to be expanded and further detailed. One critical, which does not impede the publication of this piece, is that the authors make no (or do not present) a sensitivity analyses based on taxon. Many species are exclusively benthic or pelagic. Authors could have selected a subset of the data to test to what extent all records of a certain species were being correctly classified as benthic or pelagic. The authors note that there are records also from planktonic/larval stages which does not reflect the adult stage of the organism. That is true, but dataset information can be used to assess which type of data the dataset was targeting. Again, this could be done for a subset of the data to provide a further assessment of the capacity of classification of this model.	Thank you very much for your thoughtful and constructive feedback. We are pleased that you found the manuscript to be a significant contribution and appreciate your recognition of the value both in the dataset and the pipeline we have developed. We have considered your comments and hope that the changes and clarifications made throughout the manuscript and in the responses below adequately address your concerns. While we acknowledge the value of a taxon-specific sensitivity analysis, a full-scale implementation across millions of records is not currently feasible due to inconsistent taxonomic resolution and metadata in OBIS. However, we have clarified these limitations and discussed the potential for targeted sensitivity assessments in future work or for specific user applications. In response to your comments on the model fitting, we have expanded the methods section to better detail the GAM fitting process, included discussion of diagnostic checks, and now provide 95% confidence intervals for the benthic thresholds in Extended Data Table 4 to better communicate model uncertainty.

	One further point that I would like to highlight is that the authors say that they are providing a dataset as one of the products of the work. This is great, and those curated datasets are always good for the community. But more important, and what I believe they should highlight more, is the pipeline. With over a million records being added monthly to OBIS, a new dataset will be needed soon. Thus, the real value is in the pipeline which will be further used in the future.	
Title	Could be changed to reflect more the core of the work: the method and the bias found. But just a suggestion	We would like to retain the broad title proposed in the original submission.
Introduction	Line 48: please, provide examples of which key drivers you are referring here.	We have added examples that make it clear as an introduction for the next sentence. Now reads: “A dataset with the power to answer fundamental questions, quantify biodiversity and characterize species ecological niches in the deep sea would need to be global in coverage, with equal stratification across key drivers of biodiversity such as latitude and depth.”
Introduction	Line 49: The wording “biologically irrelevant” may be wrong here. Think, for example, on pressure differences created by depth and how this affects life.	Whilst we understand what the reviewer means, their example identifies pressure as the driver (not depth). Depth and latitude are both man-made constructs. We have amended the sentence to make this clearer. Now reads: “Whilst latitude and depth are, themselves, biologically irrelevant, they serve as proxies for several other variables (e.g., pressure, temperature) and thus act as powerful predictors of biodiversity across ecosystems, including in the deep ocean.”
Introduction	Line 74-75: that is true, but it should be noted that records, in general, contains extensive additional information attached to it that could also be used to disentangle that.	Whilst it is true that some records in open-access databases like OBIS include additional information to contextualise the record, this is not true for the vast majority of historical records. Given we wanted to obtain an all-encompassing view on where has and has not been sampled, we could not

		discriminate based on fields that were only populated for the minority of the data.
Introduction	Line 81-83: this needs rephrasing; OBIS and GBIF are global data aggregators, unifying multiple data collection efforts, including a wide range of methods and initial research objectives. Both are already serving to answer fundamental questions, including on deep sea. But, you are now clearly showing that we have a pathway to improve data for the deep sea, what is extremely important.	We thank the reviewer for their comment, and did not intend to undermine the value of OBIS and GBIF. We have amended this section to incorporate the reviewer's suggestions. Now reads: "In this study, we address and rectify this gap, providing an automated solution to benthic-pelagic classification of open-access biological data. While global data aggregators such as OBIS and GBIF already support a wide range of fundamental research, our curated global dataset offers a pathway to improve the habitat-specific accuracy of these records, helping to better answer fundamental questions about life in the deep ocean."
Introduction	Line 81: maybe classification instead of filtering?	Now reads: "In this study, we address and rectify this gap, providing an automated solution to benthic-pelagic classification of open-access biological data."
Introduction	Lines 89-91: this could be more clearly explained here. Reading the rest of the methods I came to understand that, but on the first reading no.	We thank the reviewer for highlighting the possible confusion caused by this sentence. We have attempted to better explain our methods. Now reads: "For the 18.9M records sitting atop the GEBCO bathymetric grid and with OBIS record depths between 30 and 10,920 m, we created a subset for model training. To do this, we selected records whose GEBCO-derived depth matched one of 14 predefined depth horizons (i.e. GEBCO depth is 100 m, 1,000 m etc.). These discrete depth layers were chosen to ensure even coverage across the bathymetric range and reduce potential biases in depth distribution during model building."
Introduction	Line 94: add the acronym (GAM)	We have added the acronym.
Introduction	Line 99: not sure if we can call this a "deep ocean" dataset; maybe it would be better to say that represent the most comprehensive classification of marine biodiversity data.	We agree with the reviewer's suggestion. Now reads: "The resultant benthic and pelagic datasets comprising 12.7 and 6.2 million records respectively, represent the most comprehensive classification of marine biodiversity data to date."

Introduction	Line 116: “this these” – revise.	Whilst sometimes considered unnatural, the current usage is grammatically correct as data are plural (i.e., these data).
Introduction	Line 157-160: the authors could expand this part by also highlighting the numbers of known species in each of those groups. Looking at WoRMS (https://www.marinespecies.org/aphia.php?p=stats), there are ~24,000 marine Chordata, while only Mollusca is > 85,000 and Arthropoda > 73,503. It would be nice to have an overview of how taxonomically biased the pelagic/benthic datasets are (here you talk only about # of records, what about the number of species represented?).	We appreciate the suggestion to compare the number of records per phylum with the number of known marine species in each group. We agree that such a comparison would provide a valuable perspective on taxonomic biases in the dataset. However, a large proportion of the OBIS records used in this study are not identified to species level, which prevents a meaningful or accurate comparison with known species richness as listed in sources such as WoRMS. This taxonomic resolution limitation is a known challenge when working with aggregated open-access biodiversity data, and it further reinforces the need for careful data curation, such as the benthic-pelagic classification approach we propose here. We have added a brief statement to acknowledge this limitation and its implications for interpreting taxonomic patterns. Now reads: “As many records are not identified to species level, it is not possible to accurately assess how well different phyla are represented in terms of species diversity.”
Introduction	Line 167: bias -> biased	We have replaced bias with biased.
Introduction	Line 175-179: the authors propose good causes here, well chosen.	We thank the reviewer for their kind comment.
Introduction	Line 187-188: indeed, there is no global product of sampling effort, but it is possible to derive metrics from OBIS in that regards. For example one can easily calculate the sampling effort as the number of unique events in a single day or month. See for example https://doi.org/10.1111/ecog.07006 (Thyrring et al. 2024). It would be interesting to derive such a metric for at least a sample of the grid cells you are analyzing, as this would enrich your discussion. If you can’t do that assessment now, or believe is out of scope, then I advise to rephrase that part, highlighting that effort could be at play here.	We thank the reviewer for this constructive comment and for drawing our attention to the Thyrring et al. (2024) study. We agree that OBIS-derived effort metrics can be informative and appreciate the value of integrating such analyses. However, we feel this level of investigation is beyond the current scope and aims of our study. We have revised the text to acknowledge the potential for deriving sampling effort proxies from OBIS and clarified that while we do not pursue this here, it is an important avenue for future work. Now reads: “As there is no standardised global product for sampling effort, it is difficult to robustly assess how variation in effort influences observed patterns, or the degree to which

		non-submission of data to platforms like OBIS may contribute to spatial biases. While recent studies have demonstrated that proxies for effort, such as the number of unique sampling events over time, can be derived from OBIS (e.g., Thyrring et al., 2024), such analyses are beyond the scope of the current study. Nevertheless, discrepancies between mapping open-access biodiversity data and literature-derived sampling effort have been noted for some large ocean areas, although overarching spatial patterns are often broadly consistent.”
Introduction	Lines 181-194: this whole paragraph is confusing. First the authors says that “the data presented here could be explained by a combination of all three of these possible explanations”. Then they say that “we feel it is unlikely that the decline in animal standing stock with depth adequately explains our observations, or that incorporating data that is absent from OBIS into our spatial analysis would significantly change our findings. We therefore conclude these environments are truly under-sampled.”. I agree that the three must be at play together in shaping the data, but the final phrases are not reflecting that.	Both reviewers highlighted the confusing nature of this paragraph. We are saying that there are three possible explanations for the patterns we see in the data (standing stock-related, missing OBIS data and true under-sampling). Whilst explanations one and two likely influence the data, we do not feel they can individually explain the trends we see. Therefore, it is likely a combination of all three reasons as to why we see the patterns we do. We have attempted to clarify our meaning through the rewording of the final phrases, as per reviewer one’s suggestion. Now reads: “The availability of the data presented here could be explained by a combination of all three. Whilst the deep sea was considered barren by some early naturalists (Forbes’ azoic hypothesis; 22), studies from the 1960s began to dismantle the idea of a ‘low diversity and biomass deep sea’ 23, 24. This said, there is a well-established decrease in animal densities and biomass with depth for both benthic 25 and pelagic taxa 26 that could contribute to our observations. As there is no standardised global product for sampling effort, it is difficult to robustly assess how variation in effort influences observed patterns, or the degree to which non-submission of data to platforms like OBIS may contribute to spatial biases. While recent studies have demonstrated that proxies for effort, such as the number of unique sampling events over time, can be derived from OBIS (e.g., Thyrring et al., 2024), such analyses are beyond the scope of the current study. Nevertheless, discrepancies between mapping open-

		access biodiversity data and literature-derived sampling effort have been noted for some large ocean areas, although overarching spatial patterns are often broadly consistent 27, 28. Whilst explanations one (decline in animal standing stock with depth) and two (missing data in OBIS) likely influence the dataset, we do not feel they are solely responsible for the patterns in the data. Therefore, explanation three, that these environments are truly under-sampled, holds true.”
Introduction	Lines 196-206: please, provide a more detailed explanation for the reader of what you mean with the “lines” and why this would be connected to pelagic sampling.	We have added additional text into the manuscript to explain our rationale. Now reads: “In particular, the presence of continuous ‘lines’ of data points, such as those visible in the Indian Ocean directly south of the Arabian Sea (Figure 2B), suggests that some pelagic records remain within the benthic dataset. These linear patterns are characteristic of data collected using towed sampling gear such as plankton trawls or midwater nets, which are commonly deployed over long distances in pelagic surveys. In contrast, benthic sampling, especially in the deep sea, typically involves discrete sampling events at specific locations (e.g., grabs, cores, ROV deployments) rather than continuous transects. Therefore, the spatial structure of these data strongly indicates a pelagic origin.”
Introduction	Lines 208-209: again, I think the authors should better explain to the user why they should have caution. That is not a problem, but the nature of OBIS, GBIF: they are global aggregators of data, collected through multiple types of methods and across many different objectives. As so, any research done with that must be done with the appropriate care. Data must be filtered and considered according to the use case.	We view the highlighted sentence as an introduction to the rest of the paragraph on cautious approaches. However, we appreciate we are not specifically talking about differences in sampling methods, research objectives etc. We have therefore used the reviewer’s comment to provide additional examples of why caution should be taken whilst retaining the original point of the paragraph. Now reads: “Despite the inevitable errors in such as vast database, open-access global data repositories like OBIS are undeniably useful but should be used with caution, not least of all because they aggregate data collected using a wide range of methods, sampling designs, and research objectives.”

Introduction	Lines 222-224: over-fitted is not a good term here. Over-fitted refers always to model fitting “too much” to the input data, thus lacking the capacity to predict to new data. Here better to say “biased towards”	We have clarified that we are indeed talking about (modelled) inferences and predictions in this sentence. Now reads: “The consequences of biases in open-access global data towards the northern hemisphere, shallow waters and vertebrates mean any modelled inferences and predictions made using these are likely over-fitted to more heavily sampled areas/taxa”.
Introduction	Lines 232-234: indeed, this is critical and an important find of this work!	We thank the reviewer again for their kind and constructive comments.
Methods	Lines 429-432: The justification for using 30m is acceptable, but I would not say the challenges of collecting data at 100m depth are the same as collecting at 1000m. This definition should also be on the main text, so readers now why <30m is being considered deep-sea by you.	We agree that this justification should be provided in the main text and have therefore moved this sentence. We do not state the challenges are the same beyond SCUBA depths, we state they are similar. Nevertheless, we acknowledge the reviewer’s concern and have added further clarification in the text. Now reads: “Although the deep sea is commonly defined as those areas deeper than 200 m, the challenges associated with collecting data deeper than standard SCUBA depths are similar (and typically increase with depth), and therefore, for the purposes of this review, we include information below 30 m.”
Methods	Lines 442-443: I could not understand what you meant here until I went to the ED. Needs rephrasing and more detail.	We have clarified the meaning of this sentence, although the reference to the extended dataset is purposefully used as the table contextualises this sentence better than words can. Unfortunately, the Nature submission order means the extended data cannot be merged with the methods. Now reads: “Records were subsetted for specific depth horizons/bands where the GEBCO depth was a set value or small range (Extended Data, Table 1).”
Methods	General: when the target is prediction, it is generally advisable to have some sort of cross-validation (including leave-one-out). Also, a model with 100% of deviance explained appears to be over-fitted and without a validation in an external dataset it is	We thank the reviewer for raising this important point. While our GAM explains 100% of the deviance, this reflects the very tight and structured relationship between bathymetric depth and the upper limit of the benthic data cloud, as one might expect. To assess model fit and potential overfitting, we used

	impossible to assess if it is correct or not (I'm definitely not saying that your model is wrong because of the 100% deviance explained). I think the authors need to provide more details about the model fitting and how model was assessed.	the check_model() function from the performance package in R, which generated a posterior predictive check and diagnostic plots for linearity and variance homogeneity. As expected for a GAM, the linearity plot showed some deviation from a flat line, which is consistent with the model's use of smooth terms to capture non-linear relationships. The homogeneity plot showed some increase in variance at lower fitted values, but residuals remained generally centred around zero and did not suggest serious violations. We have also added an additional table into the Extended Data section, showing the 95% confidence intervals for the predicted upper limit of the benthic data cloud at the 14 predetermined depth horizons.
--	---	---

Reviewer #3

Section	Reviewer's comment	Author's response
General comment	The authors provide an automated data pipeline for extracting and separating benthic from pelagic data in open-access repositories. The author's results showed that large spatial and taxonomic biases exist in both the benthic and pelagic global datasets, favouring depths shallower than ~100 m, the northern hemisphere, and vertebrate species. The authors suggested a newly compiled, cleaned, and classified dataset to be used to identify areas of chronic under sampling and high priority regions for exploration. They recommended a coordinated strategic prioritisation of sampling as needed approach to support modelling and prediction, better management and understanding of marine life. The introduction, methodology, results, discussion, and conclusion are generally sufficiently presented. Indeed, the manuscript needs major changes before being accepted. Details are listed below:	Thank you very much for your positive assessment of our manuscript and for recognizing the importance of the automated data pipeline and its application to highlight global biases in marine biodiversity records. We appreciate your comments that the introduction, methodology, results, discussion, and conclusion are generally sufficiently presented. We acknowledge your recommendation for major revisions and have addressed all specific points raised below. In particular, we have worked to improve the clarity and detail in our methods, expanded our presentation of the modelling approach, and included additional outputs (such as confidence intervals) to better communicate model performance and uncertainty. We hope that these revisions address your concerns and strengthen the manuscript. Please find detailed responses to each of your comments below.

Abstract	Lines 6, 11: please check the author guidelines, usually reference are not allowed in the abstract as it should be a self-contained summary of author's research	We thank the reviewer for their suggestion and have removed the references from the abstract. We apologise for this – it was a hangover from a previous submission to another Nature family journal.
Introduction	Lines 22-23: we had a huge advances in technologies and exploration in the last decades, and it is probably to extreme this sentence without any recent references. Please rephrase and/or add recent references	Whilst we agree that technological advancements have allowed further exploration of the deep sea, it remains the least studied ecosystem on Earth with less than 0.0001% of areas below 200 m investigated (Danovaro et al., 2017). This is largely due to technological challenges associated with collecting data in harsh conditions. We have added recent citations to the sentence. Now reads: “Exploration of marine ecosystems beyond recreational diver depths (30m) poses a formidable challenge despite technological advancements (Feng et al., 2022), rendering these systems among the least studied on Earth (Danovaro et al., 2017).” Danovaro, R., Corinaldesi, C., Dell’Anno, A. and Snelgrove, P.V., 2017. The deep-sea under global change. Current Biology, 27(11), pp.R461-R465.
Introduction	Lines 36-41: the role of data is already clear, maybe authors wanted to convey a different message. I would suggest to clarify and add details if necessary	The purpose of this sentence is to contextualise the importance of this dataset in the policy realm and therefore we would prefer to leave it as per the original manuscript. Given we have referenced the appropriate legal instruments, we are unclear what the reviewer means by clarify and add details.
Introduction	Lines 48-49: please develop more the sentence explaining the reason why you consider “latitude and depth biologically irrelevant”	Depth and latitude are both man-made constructs; the true drivers of biodiversity are, for example, pressure and temperature, that are correlated with depth and latitude and thus serve as proxies. We have amended the sentence to make this clearer (hopefully satisfying both reviewers). Now reads: “Whilst latitude and depth are, themselves, biologically irrelevant, they serve as proxies for several other variables (e.g., pressure, temperature) and thus act as

		powerful predictors of biodiversity across ecosystems, including in the deep ocean.”
Introduction	Lines 52-53: please correct: temperature typically decrease with increasing depth	We thank the reviewer for picking up this error and have now corrected the text.
Introduction	Lines 53-55: the sentence seem to be repeated. Please rephrase it correctly.	The first part of the sentence refers to thermal energy, whilst the second refers to chemical energy.
Introduction	Lines 78-80: please remove or move to introduction, and add the correct information. These are no information related to Data sources and interpretation of the indicators, as indicated in the subtitle.	Lines 78-80 are a paragraph break and already in the introduction. We are unsure what the reviewer means.
Introduction	Lines 55-58: not very clear the message the authors want to convey. Please clarify	This sentence refers to the need for a comprehensive dataset to comprise records from multiple taxonomic groups so it's not biased towards any specific phylum, class etc. This is important as different taxonomic groups have different life histories and evolutionary strategies that would make them vulnerable to different threats.
Introduction	Line 64: please correct the web address	We thank the reviewer for picking up this error and have now corrected the text.
Introduction	Lines 64-69: OBIS and GBIF are interconnected in the last years, so I believe that it is not very useful the comparison anymore. I would suggest removing or rephrasing it	Whilst we acknowledged that the databases are connected, we anticipated a question readers may have is why we chose OBIS rather than GBIF as our test repository for this study. This sentence justifies our choice and therefore we would like to leave it in the text.
Introduction	Line 77-78: I would suggest developing more why the comparison horizontal vs vertical is a key consideration.	We have added an additional sentence at the end of this paragraph outlining the importance of this difference. Now reads: “This distinction is particularly important because pelagic species, with greater horizontal mobility, may be able to track shifting environmental conditions more readily than benthic organisms, which are often sessile or have limited vertical and horizontal dispersal capacity. As a result, their vulnerability and ecological responses to threats can differ markedly.”
Methods	Lines 86-99: please add the level of uncertainty of the automated solution.	Thank you for your helpful suggestion. Adding the 95% confidence intervals to Figure 1 is not particularly helpful given they are reasonably tight around the line given the large

		range of the y axis. Instead, we have now included 95% confidence intervals for the predicted benthic thresholds as a new Extended Data item (Table 4) to better communicate the level of uncertainty.
Methods	Figure 1: please increase font size, not readable	We have increased the font size.
Results	Lines 109-132: the variability could be strongly due to users, which decide what and where submit the data. Please, consider this aspect in the section	The section the reviewer is referring to here (patterns in the data) is simply a descriptive section based on the maps. Challenges associated with unsubmitted data are discussed in detail in the discussion. We feel this structure is appropriate, and discussing this here would lead to repetition.
Results	Figure 2: please increase font size, not readable. In addition consider to use colour blind friendly palette	We have increased the font size but prefer to remain with the original colour scheme.
Results	Line 156: please develop more the last part of the sentence otherwise remove it since it not scientifically demonstrate. In addition, “popularity”? I would suggest using a different term	We have developed the last part of the sentence outlining why the popularity of trawl sampling would lead to higher number of Chordate records. Now reads: “Distribution of records across major phyla reveal a distinct bias towards Chordata in both the pelagic and benthic datasets, perhaps unsurprising given the popularity of trawl sampling as a sampling methodology that naturally targets fish species.”
Results	Figure 3-4: please increase font size, not readable	We have increased the font size.
Discussion	Lines 171-173: please develop more the reason why your data are in contrast to Webb’ suggestion	We have added additional text to the sentence to clarify the point. Now reads: “Whilst our findings lend support to [15]’s suggestion that midwater habitats are under-recorded relative to surface waters and the seabed in terms of the number of records, our data show that this is not true for spatial coverage, with the geographic distribution of benthic records being significantly more constrained than that of the pelagic data (Figure 2B versus Figure 2C)”.
Discussion	Lines 191-194 vs 181.182: please read again and correct	Both reviewers highlighted the confusing nature of this paragraph. We are saying that there are three possible explanations for the patterns we see in the data (standing stock-related, missing OBIS data and true under-sampling). Whilst explanations one and two likely influence the data, we

		do not feel they can individually explain the trends we see. Therefore, it is likely a combination of all three reasons as to why we see the patterns we do. We have attempted to clarify our meaning through the rewording of the final phrases, as per reviewer one's suggestion. Now reads: "The availability of the data presented here could be explained by a combination of all three. Whilst the deep sea was considered barren by some early naturalists (Forbes' azoic hypothesis; 22), studies from the 1960s began to dismantle the idea of a 'low diversity and biomass deep sea' 23, 24. This said, there is a well-established decrease in animal densities and biomass with depth for both benthic 25 and pelagic taxa 26 that could contribute to our observations. As there is no standardised global product for sampling effort, it is difficult to robustly assess how variation in effort influences observed patterns, or the degree to which non-submission of data to platforms like OBIS may contribute to spatial biases. While recent studies have demonstrated that proxies for effort, such as the number of unique sampling events over time, can be derived from OBIS (e.g., Thyrring et al., 2024), such analyses are beyond the scope of the current study. Nevertheless, discrepancies between mapping open-access biodiversity data and literature-derived sampling effort have been noted for some large ocean areas, although overarching spatial patterns are often broadly consistent 27, 28. Whilst explanations one (decline in animal standing stock with depth) and two (missing data in OBIS) likely influence the dataset, we do not feel they are solely responsible for the patterns in the data. Therefore, explanation three, that these environments are truly under-sampled, holds true."
Discussion	Lines 199-201: due to incorrect logging or OBIS submission form	We agree with the reviewer as explained in the manuscript.